# Response of the Radial Growth of Woody Plants in the West Siberian Plain and Adjacent Mountainous Territories to the Characteristics of the Snow Cover

**Nikolay I. Bykov** *[ORCID], **Anna A. Shigimaga** and **Natalia V. Rygalova**

Institute for Water and Environmental Problems, Siberian Branch, Russian Academy of Sciences, 1 Molodezhnaya, Barnaul 656038, Russia

\* Correspondence: nikolai_bykov@mail.ru

**Abstract:** The dependence of the width of annual rings of woody plants on the characteristics of the snow cover is analyzed in various natural zones of the West Siberian Plain and adjacent mountainous areas: the maximum depth and water reserve for the entire winter period and for individual months, the dates of disappearing, establishment, and duration of the occurrence of a stable snow cover. It has been shown that the role of the depth and water content of snow cover for the radial growth of trees is differentiated by geographical location. On the plain, it intensifies in the forest-tundra and dry steppe. The response of radial growth to snow cover in the upper and lower parts of the forest belt is often the opposite. Dates of establishment of stable snow cover are more important for tree growth compared to dates of disappearance. Dates of disappearance of stable snow cover are more significant in the southern regions than in the northern ones. The value of the duration of the period with stable snow cover for tree growth is higher in the southern regions.

**Keywords:** West Siberian Plain; Polar Urals; Altai; Salair Ridge; snow cover; woody plants; growth rings; tree-ring width

## 1. Introduction

Snow cover is an important element of ecosystems that affects their dynamics and productivity [1]. This effect is carried out through a change in the thermal and water regime. In winter, for example, it protects plants from freezing (especially renewal buds) and wind drying. In spring, the disappearance of the snow cover controls the dates of the onset of phenophases and thus, to some extent, the duration of the growing season. The water reserves in the snow cover determine the moisture content of the soil during the growing season and, consequently, the productivity of plants. In addition, snow cover can have a mechanical effect on plants due to its sliding downslope (excluding avalanches). The effects of snow cover on plants occur in combination with other factors. At the same time, the study of the relationships in the vegetation-snow cover system makes it possible to understand the geographical differentiation of the role of snow cover as an ecological factor, as well as the possibilities of tree-ring indication for the reconstruction of snow-cover data.

Many researchers have been studying the influence of snow cover indicators on the growth of woody plants in various habitats [2–7]. The conclusions formulated in these works boil down to the fact that the influence of snow cover largely depends on geographical location [8]. The strengthening of this influence is noted in areas of prolonged occurrence of snow cover [9–12]. Based on the established dependencies, some researchers reconstructed a number of characteristics of the snow cover (maximum thickness, duration of occurrence, date of disappearance) [13–15] as well as the values of snow reserves in the river basin [16].

At the same time, there has been practically no analysis of the effect of snow cover within large territories (physico-geographical countries) characterized by the differences in

the ecological significance of snow cover. At the same time, it is known that the productivity of woody plants on the northern and upper boundaries of the forest is controlled by the sum of positive air temperatures of the growing season or its individual months [17,18]; on the southern boundary of boreal forests, the main limiting growth factor is the moistening of the territory [19]. This led to the purpose of this work—to establish the role of various characteristics of snow cover on the width of annual rings in the contrasting conditions of the West Siberian Plain and adjacent mountainous areas. The hypothesis of the study lies in the assumption that the increase in the impact of snow cover on the radial growth of woody plants occurs at the limits of forest distribution, especially in the northern and upper ones. At the same time, depending on geographic location, the set of the most significant indicators of snow cover changes.

## 2. Materials and Methods

Both natural zones and subzones of the West Siberian Plain (forest-tundra, northern taiga, middle taiga, northern and southern forest-steppe, as well as dry steppe) and the extreme positions of the forest belt (upper and lower parts) in the Polar Urals, Salair Ridge, and Altai were selected as research areas (Figure 1, Tables 1 and 2). Dendrochronological samples were taken at different times, which created a different comparison period, as well as from different species. In some areas, samples were taken from sites differing in exposure position or position within the landscape catena (Tables 1 and 2). Sampling, both in the forest-tundra and in other territories, took place in accordance with the recommendations of dendroclimatic work [20]. At each site, 30 cores were obtained from 15 trees of each studied species. Measurements of the width of the annual rings were carried out on a semi-automatic Lintab 6 installation with an accuracy of 0.01 mm. Standardization and generalization of dendrochronological series were performed in the ARSTAN program. The RBAR (Running correlation between series) and the EPS (Expressed Population Signal) were used to estimate the tree-ring chronologies for particular sites. Mean sensitivity was employed to evaluate local chronologies [21] the construction of them was carried out if the EPS value was equal to or higher than 0.85. For dendroclimatic analysis, local chronologies were used if the mean sensitivity was equal to or higher than 0.2. The COFECHA software was applied to identify missing and false tree rings. Some results of data analysis of dendrochronological samples have been published [3,4,19,22].

In the Polar Urals (Section 1 in Figure 1), samples were taken on the slopes of three exposures (Table 2) in the basin of the Bolshaya Khadata River. In the forest-tundra, samples were taken within the landscape catena at different levels (sites 3.1–3.4). However, in this paper, we consider samples obtained only on upland surfaces and in the lower parts of slopes (transit-accumulative parts of the catena). In the northern taiga, the collection sites were located in the upper (transit-eluvial) and lower (transit-accumulation) parts of the landscape catena (sites 2.1–2.6). In the middle taiga, the collection sites were located on the terrace of the Ob River (sites 4.1 and 4.2). In the forest-steppe, the collection sites were located in pine forests, while in the southern forest-steppe, they were located exclusively at the bottom of ancient runoff hollows (plots 9–12). In the dry steppe, in addition to pine forests, including those at the bottom of ancient runoff troughs, samples were also taken in field-protective forest belts (sites 5.1, 5.2, 7.2). At the same time, sites for a selection of dendrochronological samples in the dry steppe were located both in humid habitats (transit-accumulative parts of landscape catenas, sites 8.1 and 8.2) and dry habitats (on sandy ridges of belt forests, sites 6 and 7.1).

In mountainous areas, samples were taken both in the upper parts of the forest belt and in the lower ones (Table 2). Since the study area has a significant length from north to south, the altitudinal position of the upper forest boundary varied from 250 to 300 m to 2200 to 2300 m above sea level. The altitudinal position of the lower parts of the forest belt changed similarly (Table 2).

The study area is extremely diverse in terms of nival conditions (Table 3). The average long-term maximum snow cover depth for the winter period of 1966–2020 varied from 8

(Kosh-Agach weather station) to 80 (Nadym weather station) cm, and the water content of the snow cover was from 21 (Kosh-Agach) to 191 (Nadym) mm. The establishment of a stable snow cover varied from October 12 (Nadym) to November 17 (Kosh-Agach), and its disappearance from March 14 (Kosh-Agach) to May 30 (Tazovsk). The minimum average long-term duration of the period with stable snow cover was observed in Kosh-Agach (121 days) and the maximum in Tazovsk (231 days). To analyze the relationship between the width of annual rings and snow cover indicators, we used data from the meteorological stations of the state meteorological network [23,24] that are closest to the sampling points (Figure 1, Tables 1 and 2). Unfortunately, sometimes the weather stations are at a significant distance from the sampling sites. So, the Salekhard meteorological station is 115–120 km away from the sites in the Polar Urals. At the same time, weather stations whose data were used to analyze the relationship with the width of annual rings were not always located in similar conditions. For example, the Tazovsk weather station, despite the fact that it was only 40 km away from the forest-tundra areas, was nevertheless actually located in the southern tundra. A more difficult task was to compare snow cover indicators with tree-ring chronologies from the upper border of the Altai forest since all weather stations here are located at the bottom of mountain valleys or basins, where the nival conditions differ significantly from the upper levels.

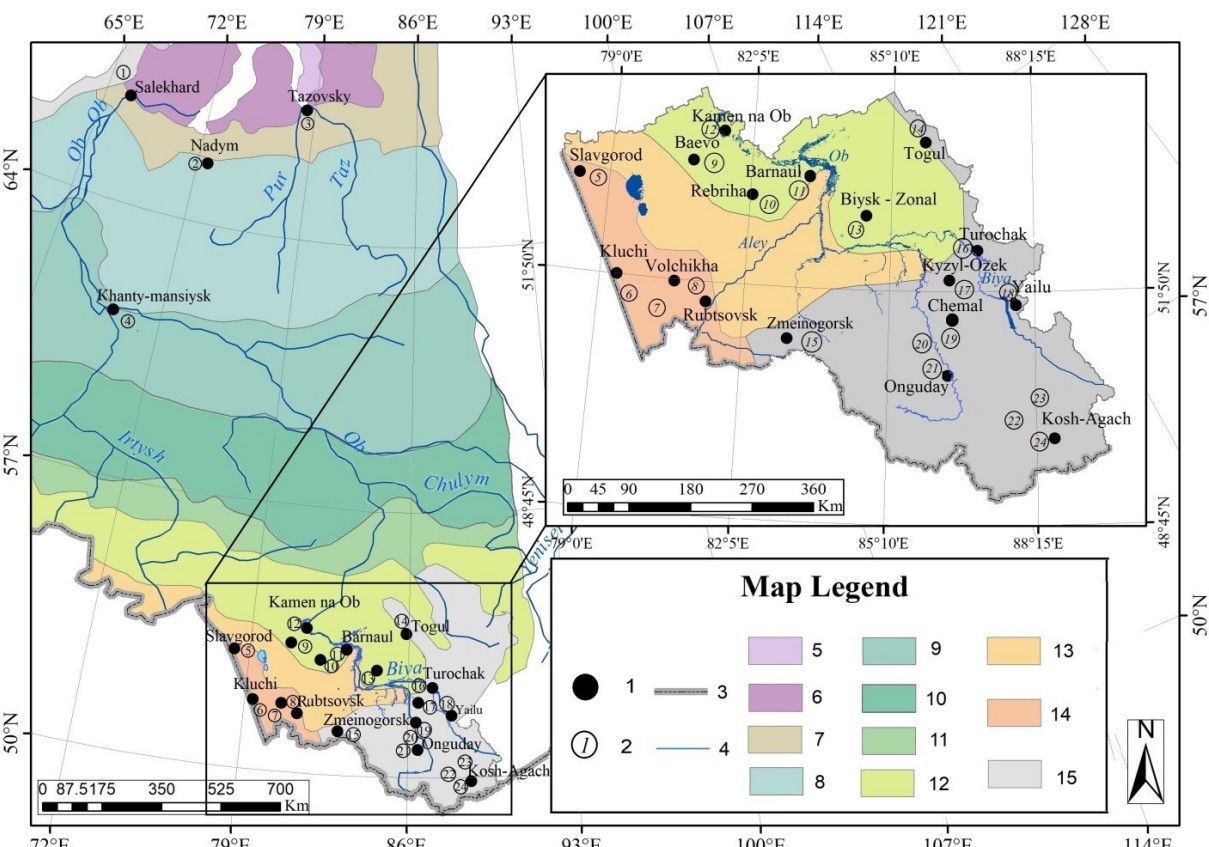

**Figure 1.** Geographical location of the sites of dendrochronological sampling: 1. Sites of the Polar Urals; 2—sites of the northern taiga; 3—forest-tundra sites; 4—sites of the middle taiga; 5–8—dry steppe sites; 9–12—sites of the southern forest-steppe; 13—site of the northern forest-steppe; 14—sites of the Salair Ridge; 15—sites of the Tigiretsky Range; 16–19, 21, 24—sites of the lower part of the forest belt; 20—site of the Seminsky ridge; 22—site of the Severo-Chuysky ridge; 23—sites of the Kurai Range. Map legend: 1—weather stations; 2—sites for the selection of dendrochronological samples; 3—state border; 4—rivers; 5—typical tundra; 6—southern tundra; 7—forest-tundra; 8—northern taiga; 9—middle taiga; 10—southern taiga; 11—mixed forests; 12—forest-steppe; 13—typical steppe; 14—dry steppe; 15—mountainous areas of altitudinal zonality.

**Table 1.** Geographical location of dendrochronological sampling sites within the West Siberian Plain.

| Lot Number | Site and Position | Species | Weather Station | Distance of the Site from the Weather Station, km |
|---|---|---|---|---|
| | forest-tundra | | | |
| 3 | 3.1. Plakor | *Larix sibirica* L. | Tazovsk | 49 |
| | 3.2. lower slope | *Larix sibirica* L. | Tazovsk | 49 |
| | 3.3. Plakor | *Larix sibirica* L. | Tazovsk | 34 |
| | 3.4. river terrace | *Larix sibirica* L. | Tazovsk | 33 |
| | northern taiga | | | |
| 2 | 2.1. lower slope | *Larix sibirica* L. | Nadym | 16 |
| | 2.2. lower slope | *Picea obovata* L. | Nadym | 16 |
| | 2.3. top of the slope | *Larix sibirica* L. | Nadym | 15 |
| | 2.4. top of the slope | *Picea obovata* L. | Nadym | 15 |
| | 2.5. top of the slope | *Pinus sibirica* | Nadym | 15 |
| | 2.6. Terrace of the Nadym River | *Pinus sylvestris* L. | Nadym | 20 |
| | Middle taiga | | | |
| 4 | 4.1. Terrace of the Ob River | *Pinus sylvestris* L. | Khanty-Mansiysk | 22 |
| | 4.2. Terrace of the Ob River | *Pinus sibirica Du Tour* | Khanty-Mansiysk | 22 |
| | dry steppe | | | |
| 5 | 5.1. forest belt | *Pinus sylvestris* L. | Slavgorod | 20 |
| | 5.2. forest belt | *Betula pendula Roth* | Slavgorod | 20 |
| 6 | pine forest, mane top, dry habitat | *Pinus sylvestris* L. | Kluchi | 17 |
| 7 | 7.1. Ancient runoff hollow, pine forest, mane top, dry habitat | *Pinus sylvestris* L. | Kluchi | 63 |
| | 7.2. forest belt | *Betula pendula Roth* | Kluchi | 43 |
| 8 | 8.1. Ancient runoff hollow, pine forest, wet habitat | *Pinus sylvestris* L. | Rubtsovsk | 72 |
| | 8.2. Ancient runoff hollow, pine forest, wet habitat | *Pinus sylvestris* L. | Volchicha | 4 |
| | southern forest-steppe | | | |
| 9 | Ancient runoff hollow, pine forest | *Pinus sylvestris* L. | Kamen' na Ob | 57 |
| 10 | Ancient runoff hollow, pine forest | *Pinus sylvestris* L. | Rebriha | 3 |
| 11 | Ancient runoff hollow, pine forest | *Pinus sylvestris* L. | Barnaul | 36 |
| 12. | Ancient runoff hollow, pine forest | *Pinus sylvestris* L. | Kamen'na Ob | 31 |
| | northern forest-steppe | | | |
| 13 | pine forest | *Pinus sylvestris* L. | Biysk-Zonal | 19 |

**Table 2.** Geographical location of sites for sampling dendrochronological samples in the mountainous territory of the basin of the Ob River.

| Lot Number | Site and Position, Height above Sea Level, in m | Species | Weather Station | Distance of the Site from the Weather Station, km |
|---|---|---|---|---|
| | Upper part of the forest belt | | | |
| 1 | 1.1. Slope southeast exposure, 250 m | *Larix sibirica* L. | Salekhard | 110 |
| | 1.2. Slope of southwestern exposure, 250 m | *Larix sibirica* L. | Salekhard | 109 |
| | 1.3. Slope with northwestern exposure, 250 m | *Larix sibirica* L. | Salekhard | 110 |
| | 1.4. Slope of southwestern exposure, 220 m | *Duschekia fruticosa Rupr.* | Salekhard | 116 |
| 15 | 15.1. Tigiretsky ridge, 1290 m | *Larix sibirica* L. | Zmeinogorsk | 56 |
| | 15.2. Tigiretsky ridge, 1430 m | *Abies sibirica* L. | Zmeinogorsk | 51 |
| 20 | Seminsky ridge, 1890 m | *Pinus sibirica Du Tour* | Biysk-Zonal | 185 |
| | | | Onguday | 47 |
| 22 | Severo-Chuysky ridge, 2280 m | *Larix sibirica* L. | Biysk-Zonal | 331 |
| | | | Onguday | 120 |
| 23 | 23.1. Kurai Range, 2120 m | *Larix sibirica* L. | Yailu | 140 |
| | Lower part of the forest belt | | | |
| 14 | 14.1. Salair Ridge, upland, 310 m | *Abies sibirica* L. | Togul | 29 |
| | 14.2. Salair Ridge, upland, 310 m | *Tilia sibirica Bayer* | Togul | 29 |
| 15 | 15.3. Tigiretsky ridge, 590 m | *Abies sibirica* L. | Zmeinogorsk | 59 |
| 16 | Terrace of the Lebed' River, 330 m | *Pinus sylvestris* L. | Turochak | 2 |
| 17 | Maima river valley, 370 m | *Pinus sylvestris* L. | Kyzyl-Ozek | 1 |
| 18 | Shore of Teletskoye Lake, 450 m | *Larix sibirica* L. | Yailu | 1 |
| 19 | Valley of the Katun River, 490 m | *Pinus sylvestris* L. | Chemal | 1 |
| 21 | Ursul basin, 1035 m | *Larix sibirica* L. | Onguday | 3 |
| 23 | 23.2 Kurai Range, 1330 m | *Larix sibirica* L. | Yailu | 134 |
| 24 | Chui hollow, 1760 m | *Larix sibirica* L. | Kosh-Agach | 14 |

During the study period, at many meteorological stations, there is a tendency to increase the depth and water content of the snow cover. At the same time, when comparing snow cover indicators for the periods 1966–2020 and 1900–2020, it is found that the difference between the values does not exceed 10%. Thus, year-to-year changes in these indicators play a more important role in establishing statistical relationships between snow cover characteristics and tree growth indices than long-term trends. In the establishment, disappearance, and duration of stable snow cover, the difference in the indicated period is also insignificant—1–3 days. The relationship between tree-ring chronologies and snow cover indicators was estimated by the Pearson correlation coefficients. The period of comparison of snow cover indicators with the dendrochronological series was determined by the time of core sampling in specific locations.

**Table 3.** Average long-term values of snow cover for the meteorological stations used for the analysis for the period 1966–2020 (Hmax is the maximum depth of snow cover for the winter period, cm; Wmax is the maximum water content of snow cover for the winter period, mm; Du is the date of establishment of stable snow cover; Dr is the date of disappearance of stable snow cover; *p* is the duration of the occurrence of stable snow cover, in days).

| Weather Station | Height above Sea Level, m | Hmax | Wmax | Du | Dr | *p* |
|---|---|---|---|---|---|---|
| Tazovsky | 26 | 35 | 93 | 10.X | 30.V | 231 |
| Salekhard | 15 | 54 | 143 | 13.X | 20.V | 218 |
| Nadym | 14 | 80 | 191 | 12.X | 15.V | 213 |
| Khanty-Mansiysk | 46 | 62 | 148 | 25.X | 23.IV | 180 |
| Slavgorod | 125 | 22 | 59 | 12.XI | 01.IV | 139 |
| Kluchi | 142 | 21 | 52 | | | |
| Volchicha | 207 | 33 | 93 | | | |
| Rubtsovsk | 216 | 21 | 49 | 09.XI | 06.IV | 141 |
| Kamen na Ob | 127 | 29 | 71 | 06.XI | 02.IV | 146 |
| Baevo | 121 | 24 | 66 | | | |
| Rebriha | 218 | 35 | 97 | 06.XI | 08.IV | 153 |
| Barnaul | 183 | 34 | 85 | 03.XI | 08.IV | 155 |
| Biysk-Zonal | 210 | 50 | 130 | 10.XI | 10.IV | 140 |
| Togul | 310 | 59 | 152 | | | |
| Zmeinogorsk | 354 | 44 | 116 | 09.XI | 09.IV | 151 |
| Turochak | 330 | 69 | 169 | 02.XI | 16.IV | 166 |
| Yailu | 450 | 53 | 122 | 12.XI | 01.IV | 139 |
| Kyzyl-Ozek | 370 | 56 | | 05.XI | 07.IV | 152 |
| Chemal | 490 | 13 | 23 | | | |
| Onguday | 1039 | 19 | 34 | | | |
| Kosh-Agach | 1760 | 8 | 21 | 17.XI | 14.III | 121 |

## 3. Results and Discussion

The northernmost natural zone of the West Siberian Plain where we carried out the research was the forest-tundra. It was found [4] that tree-ring chronologies here are distinguished by a high value of the expressed population signal (EPS) (0.90–0.96) and a mean sensitivity (0.33–0.43) (Table 4). June (R = 0.32–0.46) and July air temperatures are the most significant for the radial growth of forest-tundra trees (R = 0.38–0.57), but for some areas, January temperature is also significant (up to R = 0.35). Winter precipitation, as a rule, has a positive effect on the growth of trees in the subsequent growing season, while summer precipitation has a negative effect. The precipitations of February (up to R = 0.39) and March (up to R = 0.31) of the current year and November of the previous year (up to R = 0.32) are most significant.

**Table 4.** Characteristics of the studied tree-ring chronologies in the natural zones of the West Siberian Plain.

| Lot Number | Site and Position | Average Age of Trees | Rbar | EPS | Mean Sensitivity | Comparison Period with Weather Data |
|---|---|---|---|---|---|---|
| | | forest-tundra | | | | |
| 3 | 3.1. Plakor | 136 | 0.37 | 0.95 | 0.40 | 1966–2020 |
| | 3.2. lower slope | 98 | 0.23 | 0.90 | 0.43 | 1966–2020 |
| | 3.3. Plakor | 77 | 0.48 | 0.96 | 0.33 | 1966–2020 |
| | 3.4. river terrace | 135 | 0.29 | 0.93 | 0.38 | 1966–2020 |
| | | northern taiga | | | | |
| 2 | 2.1. lower slope | 119 | 0.32 | 0.94 | 0.33 | 1966–2020 |
| | 2.2. lower slope | 105 | 0.22 | 0.89 | 0.29 | 1966–2020 |
| | 2.3. top of the slope | 54 | 0.73 | 0.96 | 0.30 | 1966–2020 |
| | 2.4. top of the slope | 103 | 0.16 | 0.85 | 0.31 | 1966–2020 |
| | 2.5. top of the slope | 139 | 0.30 | 0.93 | 0.32 | 1966–2020 |
| | 2.6. Terrace of the Nadym River | 188 | 0.49 | 0.97 | 0.34 | 1966– 2020 |
| | | Middle taiga | | | | |
| 4 | 4.1. Terrace of the Ob River | 104 | 0.49 | 0.97 | 0.26 | 1966–2020 |
| | 4.2. Terrace of the Ob River | 124 | 0.64 | 0.98 | 0.26 | 1966–2020 |
| | | dry steppe | | | | |
| 5 | 5.1. forest belt | 41 | 0.67 | 0.98 | 0.22 | 1975–2018 |
| | 5.2. forest belt | 42 | 0.64 | 0.98 | 0.28 | 1978–2018 |
| 6 | pine forest, mane top, dry habitat | 150 | 0.47 | 0.96 | 0.28 | 1966–2007 |
| 7 | 7.1. Ancient runoff hollow, pine forest, mane top, dry habitat | 186 | 0.45 | 0.96 | 0.33 | 1966–2007 |
| | 7.2. forest belt | 39 | 0.63 | 0.98 | 0.33 | 1966–2018 |
| 8 | 8.1. Ancient runoff hollow, pine forest, wet habitat | 120 | 0.48 | 0.96 | 0.3 | 1966–2005 |
| | 8.2. Ancient runoff hollow, pine forest, wet habitat | 127 | 0.44 | 0.96 | 0.31 | 1966–2002 |
| | | southern forest-steppe | | | | |
| 9 | Ancient runoff hollow, pine forest | 135 | 0.5 | 0.97 | 0.32 | 1966–2005 |
| 10 | Ancient runoff hollow, pine forest | 124 | 0.38 | 0.95 | 0.25 | 1966–2007 |
| 11 | Ancient runoff hollow, pine forest | 124 | 0.46 | 0.96 | 0.25 | 1966–2007 |
| 12 | Ancient runoff hollow, pine forest | 187 | 0.39 | 0.95 | 0.27 | 1966–2007 |
| | | northern forest-steppe | | | | |
| 13 | pine forest | 118 | 0.54 | 0.97 | 0.3 | 1966–2020 |

Analysis of the relationship of snow cover indicators with the width of annual rings in the forest-tundra showed that the reaction of woody plants depends on the local position of trees within the landscape catena. Larch trees react positively to the maximum values of depth (R = 0.23–0.58) and water reserve of snow cover (R = 0.43–0.60) for the winter period. At the same time, the indicators of snow cover depth on the snow course survey are more significant (R = 0.40–0.58) than on the weather site (R = 0.23–0.49) (Figure 2). Statistically significant correlation coefficients ($p < 0.05$) for individual months are observed from November to April (Table 5), reaching values from 0.25 to 0.61. Probably, such a reaction of trees is due to the fact that an increase in the depth of the snow cover enhances its heat-insulating properties, resulting in less freezing of soils, thus contributing to their faster warming up during the warm season. A similar reaction was noted in the northern taiga of Yakutia by other authors [25]. Thus, the hypothesis that an increase in the depth of the snow cover in the northern regions causes a decrease in radial growth in the subsequent growing season due to a reduction in the growing season due to its later melting [15] is not confirmed by our studies. Correlation analysis of the dates of the disappearance of a stable snow cover with its maximum depth and water reserve in winter indicates the absence of a statistically significant relationship ($p < 0.05$) between these indicators (Figure 2).

**Table 5.** Coefficients of correlation between the width of annual rings of woody plants in the studied sites (Figure 1, Tables 1 and 2) with the maximum depth/water content of snow cover for the months of the winter period on snow course survey for the period 1966–2020. (Statistically significant correlation coefficients are in bold at $p < 0.05$).

| Site | October | November | December | January | February | March | April | May |
|---|---|---|---|---|---|---|---|---|
| 1.1 | **0.25/0.30** | **0.25/0.24** | **0.34/0.36** | **0.22/0.25** | **0.34/0.28** | **0.33/0.33** | 0.16/0.16 | 0.04/0.10 |
| 1.2 | **0.30/0.31** | **0.32/0.23** | **0.39/0.37** | **0.28/0.29** | **0.42/0.34** | **0.41/0.39** | 0.21/0.19 | 0.00/0.03 |
| 1.3 | **0.31/0.28** | **0.25**/0.18 | **0.35/0.29** | **0.27/0.24** | **0.41/0.30** | **0.34/0.29** | 0.19/0.14 | 0.06/0.07 |
| 1.4 | **0.40/0.40** | 0.24/0.13 | **0.33/0.28** | **0.32/0.28** | **0.31**/0.22 | **0.27/0.25** | 0.17/0.06 | 0.06/0.02 |
| 2.1 | 0.19/**−0.32** | 0.06/0.00 | 0.11/0.02 | −0.01/−0.06 | 0.04/−0.05 | −0.06/−0.14 | 0.00/−0.09 | |
| 2.2. | 0.16/−0.20 | −0.07/−0.06 | −0.07/−0.01 | **−0.23**/−0.22 | **−0.27/−0.28** | **−0.23/−0.32** | −0.10/−0.21 | |
| 2.3 | 0.20/−0.14 | 0.06/−0.01 | 0.14/0.05 | 0.06/0.02 | 0.13/0.06 | 0.00/−0.11 | −0.01/−0.04 | |
| 2.4 | **0.31**/−0.08 | 0.08/0.08 | 0.01/0.07 | −0.13/−0.02 | −0.15/−0.08 | **−0.26/−0.28** | −0.18/−0.20 | |
| 2.5 | −0.04/**−0.33** | −0.08/−0.13 | 0.04/−0.01 | −0.07/−0.13 | −0.05/−0.17 | −0.04/−0.15 | 0.15/0.09 | |
| 2.6 | −0.16/−0.01 | −0.16/−0.19 | −0.07/−0.11 | −0.10/−014 | −0.10/−0.19 | −0.09/−0.07 | −0.02/−0.12 | |
| 3.1 | 0.09/0.14 | **0.33/0.28** | **0.42/0.42** | **0.43/0.37** | **0.37/0.42** | **0.42/0.42** | **0.48/0.37** | 0.17/0.12 |
| 3.2 | 0.06/0.16 | **0.32/0.37** | **0.43/0.44** | **0.45/0.39** | **0.40/0.42** | **0.43/0.43** | **0.47/0.39** | 0.19/0.22 |
| 3.3 | 0.12/0.09 | **0.50/0.25** | **0.59/0.47** | **0.61/0.45** | **0.56/0.52** | **0.53/0.50** | **0.54/0.41** | 0.15/0.11 |
| 3.4 | 0.22/0.19 | **0.51/0.35** | **0.57/0.47** | **0.60/0.47** | **0.56/0.57** | **0.55/0.57** | **0.61/0.48** | **0.30**/0.21 |
| 4.1 | −0.06/0.12 | **−0.30/−0.39** | −0.14/**−0.38** | −0.06/−0.21 | 0.02/−0.22 | −0.17/**−0.29** | −0.17/−0.15 | 0.00/0.15 |
| 4.2 | 0.02/0.03 | −0.15/**−0.37** | 0.08/−0.21 | **0.29**/−0.14 | **0.24**/−0.07 | 0.05/−0.21 | 0.14/−0.06 | 0.16/**0.39** |
| 5.1. | 0.28/0.18 | 0.13/0.08 | **0.31/0.26** | 0.20/0.17 | 0.13/0.13 | 0.06/0.08 | 0.55/0.17 | |
| 5.2 | 0.27/**0.74** | 0.09/0.03 | **0.29**/0.21 | 0.22/0.17 | 0.16/0.10 | 0.05/0.02 | 0.32/−0.04 | |
| 6 | 0.05/−0.23 | 0.03/0.09 | 0.15/**0.36** | 0.13/0.16 | 0.13/0.08 | 0.12/0.15 | 0.29/0.09 | |
| 7.1 | −0.13/**−0.56** | 0.21/**0.27** | 0.16/**0.39** | 0.09/0.10 | 0.09/0.06 | 0.12/0.12 | 0.32/0.24 | |
| 7.2 | 0.27/0.06 | **0.38/0.33** | **0.36/0.50** | **0.34/0.46** | **0.27/0.26** | 0.15/**0.29** | −0.18/−0.28 | |
| 8.1 | −0.24 | **−0.37/−0.47** | −0.17/−0.26 | **−0.39/−0.42** | **−0.31/−0.39** | −0.14/−0.12 | −0.32/−0.51 | |
| 8.2 | −0.37/−0.50 | **−0.82**/−0.16 | **−0.76**/−0.15 | **−0.40**/−0.13 | **−0.65**/−0.10 | 0.05/−0.11 | −0.12/−0.12 | |
| 9 | −0.17/**−0.69** | −0.01/−0.13 | 0.02/0.05 | −0.06/−0.09 | −0.04/−0.10 | −0.15/−0.17 | −0.25/−0.24 | |

**Table 5.** *Cont.*

| Site | October | November | December | January | February | March | April | May |
|---|---|---|---|---|---|---|---|---|
| 10 | −0.17/−0.31 | 0.00/**−0.27** | 0.08/0.08 | −0.07/−0.14 | 0.05/0.08 | −0.01/0.02 | −0.12/−0.10 | |
| 11 | **−0.36/−0.49** | −0.03/**−0.31** | 0.08/−0.03 | −0.02/−0.16 | 0.09/0.06 | 0.10/−0.12 | −0.17/**−0.30** | |
| 12 | −0.03/**−0.66** | −0.10/−0.24 | **−0.26/−0.29** | **−0.26**/−0.24 | 0.04/0.07 | 0.06/0.09 | −0.12/−0.05 | |
| 13 | −0.06/0.03 | −0.03/−0.04 | **−0.39/−0.34** | **−0.29/−0.31** | **−0.34/−0.28** | **−0.34/−0.31** | −0.21/0.03 | |
| 14.1 | −0.04/−0.04 | −0.19/**−0.31** | −0.06/−0.17 | −0.07/−0.04 | −0.05/−0.03 | −0.15/−0.10 | −0.22/−0.21 | |
| 14.2 | **−0.30**/−0.19 | −0.20/−0.16 | −0.01/−0.08 | 0.16/0.14 | 0.05/0.03 | 0.04/0.04 | 0.15/0.18 | |
| 15.1 | −0.21/0.14 | 0.04/−0.08 | 0.13/0.10 | 0.05/0.15 | 0.11/0.14 | 0.09/0.14 | 0.13/0.17 | |
| 15.2 | **−0.31**/0.12 | −0.02/−0.11 | 0.10/0.09 | 0.00/0.12 | 0.05/0.11 | 0.04/0.12 | 0.11/0.19 | |
| 15.3 | **0.31**/0.03 | 0.06/−0.05 | 0.02/−0.12 | 0.10/−0.12 | −0.04/−0.13 | −0.08/−0.19 | −0.11/**−0.26** | |
| 16 | −0.15/0.11 | −0.03/0.04 | −0.24/−0.15 | −0.10/−0.04 | −0.05/−0.01 | −0.21/−0.20 | −0.10/−0.07 | |
| 17 | 0.00/ | −0.01/ | −0.05/ | 0.10/ | 0.07/ | 0.19/ | −0.26/ | |
| 18 | −0.17/−0.22 | 0.18/0.07 | 0.08/0.13 | 0.18/0.07 | **0.33**/0.20 | 0.38/0.29 | 0.38/0.55 | |
| 19 | 0.07/−0.41 | 0.01/0.08 | 0.16/0.12 | 0.21/0.22 | **0.47/0.41** | **0.34/0.41** | −0.12/**−0.41** | |
| 20 | **0.35**/0.21 | 0.17/0.01 | 0.22/0.20 | 0.19/**0.25** | 0.24/0.25 | 0.31/0.33 | 0.37/0.34 | |
| 20 | −0.32/**−0.74** | 0.10/0.21 | 0.17/0.12 | 0.09/0.14 | 0.14/0.10 | **0.23**/0.13 | −0.07/**0.83** | |
| 21 | 0.29/−0.29 | −0.08/−0.13 | −0.04/−0.10 | −0.09/−0.13 | −0.09/−0.08 | 0.05/−0.03 | 0.20/**0.49** | |
| 22 | −0.03/0.18 | 0.07/0.17 | −0.09/0.12 | −0.08/**0.23** | 0.01/**0.25** | 0.08/**0.25** | 0.06 | |
| 22 | **0.38/0.45** | 0.04/0.02 | 0.17/0.09 | 0.03/−0.01 | 0.05/0.09 | 0.12/0.11 | **0.36/0.34** | |
| 23.1 | **−0.61**/0.10 | −0.23/**−0.32** | −0.22/**−0.26** | −0.22/−0.20 | **−0.25**/−0.12 | **−0.36**/−0.22 | −0.29/−049 | |
| 23.2 | **0.68**/−0.08 | 0.00/0.02 | 0.18/0.21 | 0.10/0.11 | 0.02/0.02 | 0.09/0.15 | 0.44/0.44 | |
| 24 | −0.19/−0.15 | 0.04/−0.02 | 0.06/0.05 | −0.01/0.00 | −0.08/−0.06 | 0.00/−0.09 | | |

Statistically significant correlation coefficients are also found between forest-tundra tree-ring chronologies and the dates of the establishment of a stable snow cover, in both this and previous years (Figure 3). The later its establishment occurs, the greater the increase in larch trees. In general, due to the trend of increasing snow reserves in this region, we should expect an increase in the growth rate of local larches.

In the northern taiga, the EPS at sites varies from 0.90 to 0.96, and the mean sensitivity of local chronologies ranges from 0.30 to 0.39, which is lower than in the forest-tundra (Table 4). At the same time, the North taiga chronologies are weakly related to air temperature [4]. Significant correlation coefficients are found between larch trees and the July temperature ($R = 0.27$–0.28, $p < 0.05$) and sometimes in June ($R = 0.33$). The response to precipitation in the North taiga chronologies is even less significant. Only the precipitation in October during the previous year (up to a maximum of $R = 0.31$) has a statistically significant correlation with the tree-ring width.

An analysis of the relationship between all studied indicators of snow cover and tree-ring width in the northern taiga showed that the effect of snow cover on the radial growth of woody plants is insignificant (Table 5 and Figure 2). Only spruces (plots 2.2 and 2.4) show a statistically significant negative correlation with the maximum depth and water content of snow cover for certain months from January to March (Table 5). The radial growth of larch (site 2.1) and pine trees (site 2.6) is also negatively affected by the water equivalent value in October of the previous year.

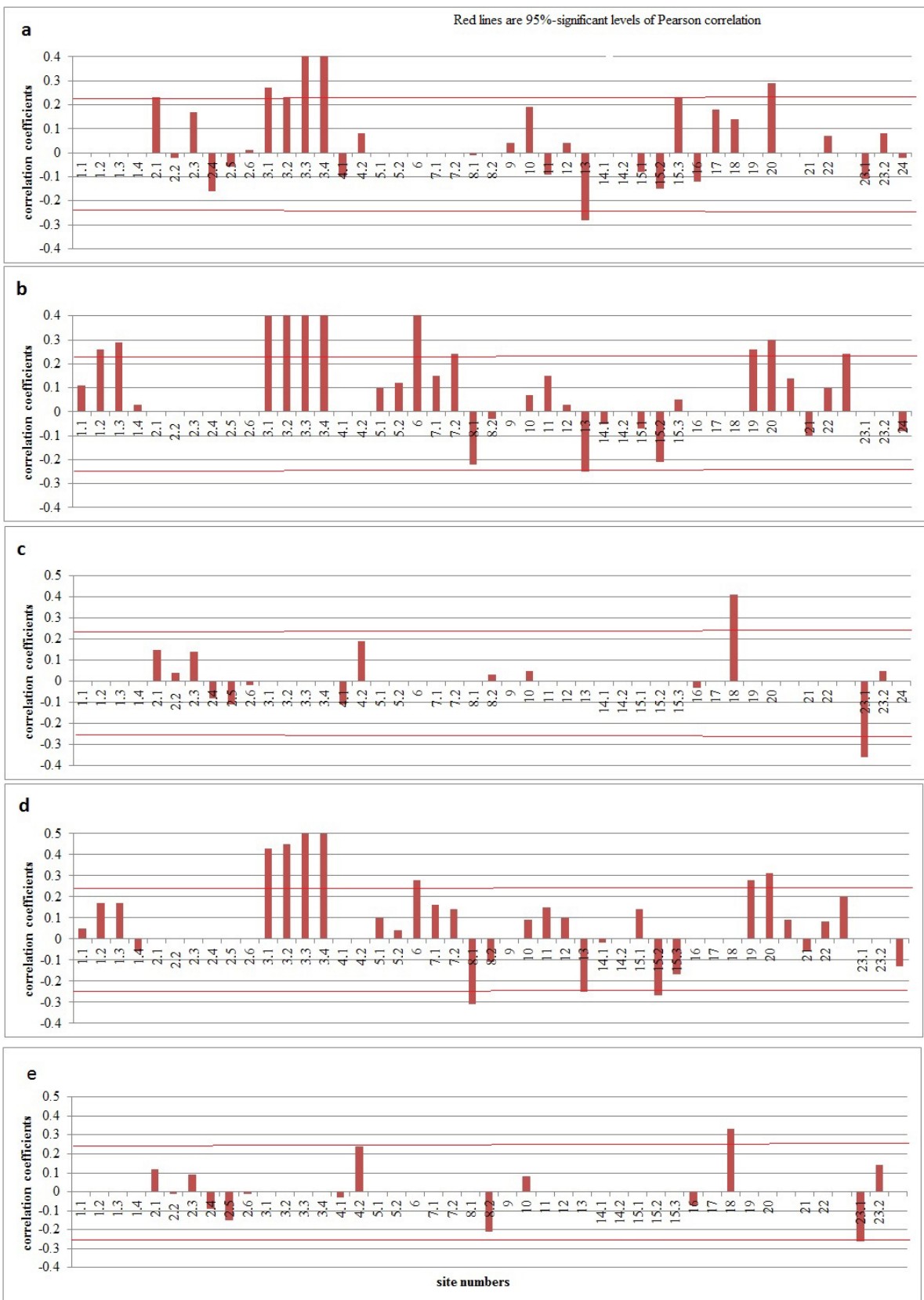

**Figure 2.** Response of tree ring width to the maximum characteristics of snow cover in the winter season for the period 1966–2020: (**a**) depth on the meteorological site, (**b**) depth on the snow course survey in the field, (**c**) depth on the snow course survey in the forest, (**d**) water storage on the snow course survey in field, (**e**) water reserve on the snow course survey in the forest.

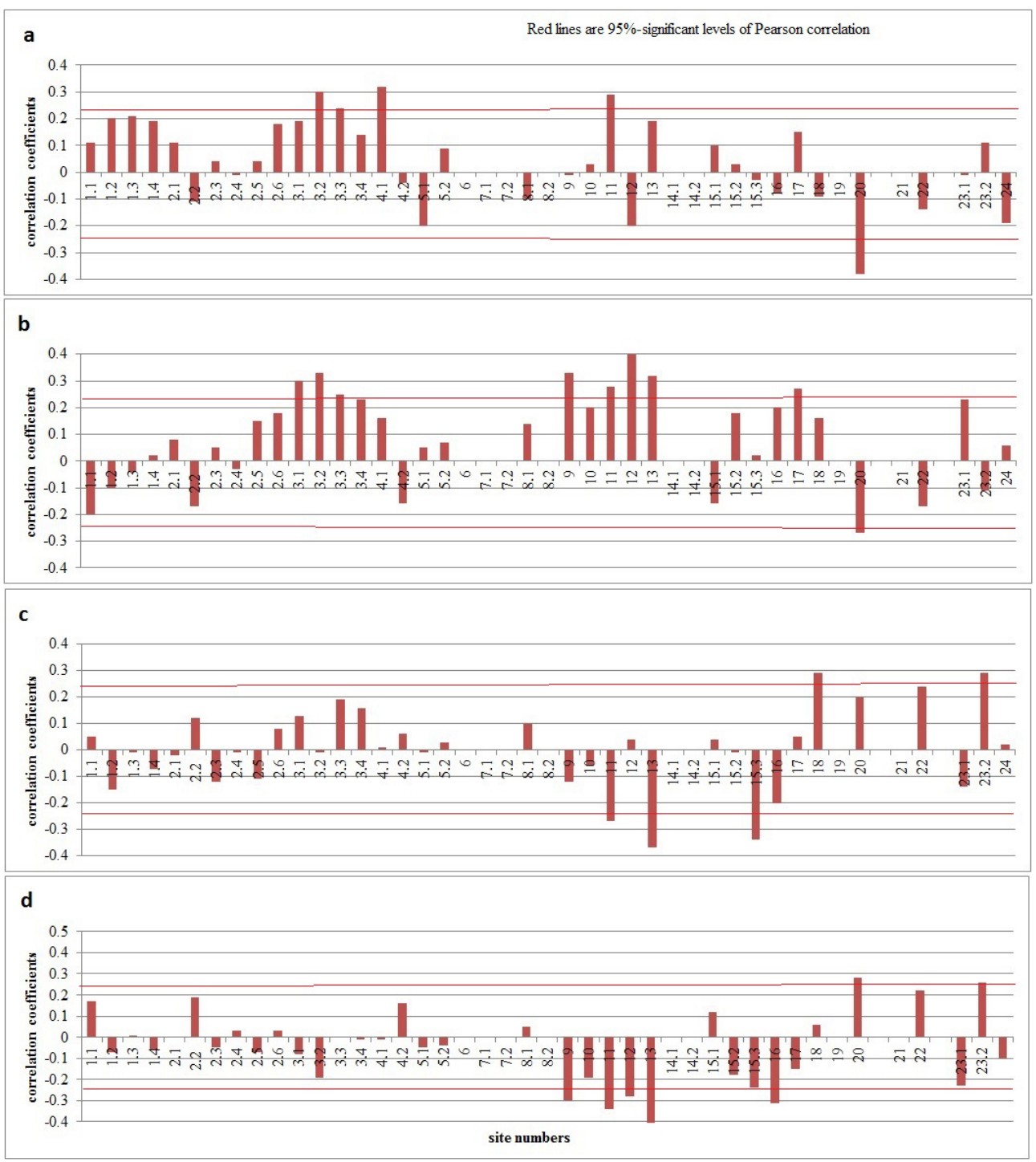

**Figure 3.** The response of the width of tree rings to the dates of establishment ((**a**)—calendar year; (**b**)—previous year), disappearance (**c**), and duration (**d**) of stable snow cover for 1966–2020.

In the middle taiga, a single population signal is 0.97 (*Pinus sylvestris* L.) and 0.98 (*Pinus sibirica Du Tour*), but the sensitivity of the local chronologies is low (0.26) (Table 4). Chronologies weakly respond to air temperature. There are statistically significant relationships between the tree-ring width of Siberian stone pine and the temperatures in April (R = −0.23) and September (R = −0.27) of the current year and for pine October (R = 0.23) during the past year and in January (R = 0.24) and April (R = −0.25) of the current year. The reaction of pine to precipitation is statistically significant only in August (R = 0.23) and in October (R = −0.29) of the prior year as well as in July of the current year (R = −0.37).

Siberian pine reacts to precipitation in August (R = 0.25) and December (R = 0.25) of the prior year and in April of the current year (R = −0.26).

Pine (*Pinus sylvestris* L.) trees (site 4.1) react negatively to an increase in the maximum depth (November) and water reserve (November, December, March) of snow cover for certain months (Table 5). At the same time, the late establishment of a stable snow cover favorably affects the tree-ring width of pines (Figure 3). By contrast, pine (*Pinus sibirica Du Tour*) (site 4.2) often reacts positively to an increase in the depth of the snow cover in January and February, as well as an increase in its water reserve in May.

In the northern forest-steppe (site 13), the single population signal is 0.97, and the mean sensitivity of the local pine chronology is 0.3 (Table 4). The influence of air temperature on the radial growth of pines is weak there. Only with the temperatures in the past October (0.25) and the current August (0.24) are there statistically significant correlation coefficients. A positive reaction of the radial growth of pine to precipitation in March (R = 0.30) and June (R = 0.23) is found. However, at the same time, pines there react negatively to an increase in the maximum depth and water content of snow cover both in individual months from December to March (Table 5) and in the winter period as a whole (Figure 2). The increase in the duration of winter and the later disappearance of stable snow cover negatively affects the tree-ring width of pines in the northern forest-steppe (Figure 3). By contrast, the later establishment of a stable snow cover in the winter preceding the growing season has a positive effect on the radial growth of pines in the northern forest-steppe.

In the southern forest-steppe, the single population signal varied in the studied sites (sites 9–12) from 0.91 to 0.95, and the mean sensitivity was in the range of 0.19 to 0.22 (Table 4). The air temperatures of the months during the warm period mainly play a negative role [4]. At the same time, the most significant role is played by the temperatures of the preceding August (R = −0.23−−0.42), as well as the temperatures in June (R = −0.34−−0.37), July (R = −0.26−−0.35), and May (up to R = −0.30) of the current year. Atmospheric precipitation, by contrast, more often has a positive effect on the radial growth of pines in the southern forest-steppe. In this case, it is not only spring-summer precipitation of the current year, especially from April to June inclusive (R = 0.26–0.42) but also precipitation in August (R = 0.28–0.37) and October (R = 0.26–0.37) of the previous year.

Snow cover on the whole acts as a negative factor for the radial growth of pines (sites 9–12) on the southern forest-steppe. The longer the winter, the earlier a stable snow cover is established and the later it disappeared, the less growth there is in pines (Table 5). Pine trees also react negatively to the maximum values of snow cover depth and water content for individual months (Table 5).

In the dry steppe (sites 5–8), a single population signal varied from 0.91 to 0.99 (for deciduous trees of forest belts), and the mean sensitivity is in the range of 0.20–0.35 (Table 4). In this subzone, the radial growth of pines is negatively affected by the temperatures of May–July in the current year (in various areas up to R = −0.30−−0.36), as well as in August and September during the past year (−0.28−−0.32). Birches (5.2 and 7.2) significantly react to the July temperature of the current year. However, atmospheric precipitation has a greater influence on the radial growth of dry steppe trees, especially pines. Most significant for pines is the amount of precipitation in August of the previous year (R = 0.50–0.54) and in May of the current year (R = 0.53–0.54). There are significant correlations between the tree-ring width of pine trees with the amount of precipitation in September of the past year as well as in April and July of the current year. Birch responds positively to the amount of precipitation of individual months of the winter and summer periods, in both the current (March, August) and past year (October, December).

In general, in the dry steppe, the relationship between snow cover and the tree-ring width is better than in the southern forest-steppe. However, the response of trees depends on the geographical location of the stands. A positive relationship with the maximum depth (R = 0.46) and water reserve (0.28) of the snow cover for the period of its maximum in winter is demonstrated by pines located on crests, that is, in dry habitats (sites 6 and 7.1), as well as by birches (R = 0.24) in protective forest strips located on the slope of the

landscape catena (platform 7.2) (Figure 2). At the same time, pines respond positively to the maximum water supply of snow cover in November and December, and birches respond positively to the maximum depth and water supply of most months from December to April (R = 0.26–0.50) (Table 5). Trees located in wet habitats (sites 8.1 and 8.2) often react negatively to maximum water reserve rates for winter (up to R = −0.31) and especially for some months in December–March (up to R = −0.31−−0.82), which is probably due to the high level of groundwater standing at these sites after snowy winters. Pines and birches in the dry steppe react poorly to the dates of establishment, disappearance, and duration of stable snow cover (Figure 3). However, previously conducted similar studies [4] on poplar showed that the longer the period of occurrence of snow cover and the later it disappears in spring, the more intense the radial growth of poplar, especially on the river-basin plain (up to R = −0.38). This can probably be explained by the fact that in the dry steppe, the late snow cover provides optimal moisture on the river basins during the period of intensive tree growth, which occurs in May and June [19], as well as in other regions [26].

The reaction of the trees of the upper forest boundary of the studied mountain areas to the indicators of snow cover is different and is determined by the species composition and geographical location. In the Polar Urals (sites 1.1–1.4), with a high EPS value (0.96–0.98) and a mean sensitivity of local chronologies (0.39–0.52) (Table 6), larch and alder respond positively to an increase in temperatures in December–February (up to R = 0.32) and especially in June and July (up to R = 0.30–0.70). Winter precipitation has a positive effect on the growth of woody plants, although there are statistically significant correlation coefficients in the rows of the width of tree rings only with December precipitation (R = 0.30–0.42). A negative reaction is noted for May precipitation (up to R = −0.30).

As a rule, the trees of the upper border of the forest of the Polar Urals react positively to the maximum values of the depth and water content of the snow cover in certain months of the winter period from October to March (Table 5), especially in February and March. The lower value of the relationship with the maximum snow cover depth for the entire winter period (Figure 2) is probably due to the fact that the maximum snow accumulation in the long term varies from month to month.

At the upper forest boundary of the Tigiretskii Range (sites 15.1 and 15.2), with a high EPS value (0.93–0.98) and a satisfactory value of the mean sensitivity (*Larix sibirica* L.—0.35; *Abies sibirica* L.—0.30) (Table 6), the reaction of tree-ring chronologies on climatic factors depend on the species. It was established that the radial growth of larches is positively affected by February precipitation (R = 0.43) and June–July temperatures (R = 0.55) and negatively by August temperatures (R = −0.45). The radial growth of firs is positively affected by the temperatures of June–July (R = 0.26), especially the latter (R = 0.53), and negatively by precipitation in May–August (R = −0.33), again especially in July (R = −0.42). Local larches of the tree-ring chronology do not react to snow cover indicators, and fir trees react negatively (R = −0.27) to the maximum water content of snow cover throughout the winter period and October (R = −0.31) (Figure 2, Table 5).

The pine (*Pinus sibirica Du Tour*) of the upper boundary of the forest of the Seminsky Range (site 20) with a high EPS (0.94) and a satisfactory mean sensitivity of local chronologies (0.29) (Table 6) respond positively to the August temperatures (R = 0.36–0.37) of the past and current year, as well as October of the earlier year (R = 0.33). Statistically significant correlation coefficients between the tree-ring width and the amount of atmospheric precipitation are available here only for September of the previous year (R = −0.25). Analysis of the relationship between tree-ring chronologies and snow cover indicators revealed that pine (*Pinus sibirica Du Tour*) chronologies respond better to the snow cover values of the Biysk-Zonalnaya meteorological station than in the closer Onguday meteorological station [3]. This is probably due to the fact that the precipitation regime at the upper forest boundary of the Seminsky Range is more synchronous with the northern regions than with the Central Altai (Onguday weather station). In general, it is a positive reaction of the radial growth of pines (*Pinus sibirica Du Tour*) to the maximum values of the depth and water content of the snow cover (R = 0.29–0.31) (Figure 2), as well as the duration of

the occurrence of a stable snow cover (R = 0.28) (Figure 3). However, the reaction to the dates of the establishment of stable snow cover of the current and previous year is negative (R = −0.38 and −0.27, respectively) (Figure 3) [3].

**Table 6.** Characteristics of the studied tree-ring chronologies in the mountainous territory of the basin of the Ob River.

| Lot Number | Site and Position, Height above Sea Level, in m | Average Age of Trees | Rbar | EPS | Mean Sensitivity | Comparison Period with Weather Data |
|---|---|---|---|---|---|---|
| | *Upper part of the forest belt* | | | | | |
| 1 | 1.1. Slope southeast exposure, 250 m | 55 | 0.60 | 0.98 | 0.40 | 1966–2020 |
| | 1.2. Slope of southwestern exposure, 250 m | 56 | 0.68 | 0.98 | 0.45 | 1966–2020 |
| | 1.3. Slope with northwestern exposure, 250 m | 115 | 0.48 | 0.96 | 0.52 | 1966–2020 |
| | 1.4. Slope of southwestern exposure, 220 m | 47 | 0.46 | 0.96 | 0.39 | 1966–2020 |
| 15 | 15.1. Tigiretsky ridge, 1290 m | 130 | 0.24 | 0.93 | 0.35 | 1966–2020 |
| | 15.2. Tigiretsky ridge, 1430 m | 55 | 0.59 | 0.98 | 0.30 | 1966–2020 |
| 20 | Seminsky ridge, 1890 m | 61 | 0.34 | 0.94 | 0.29 | 1966–2020 |
| | | | | | | 1966–2020 |
| 22 | Severo-Chuysky ridge, 2280 m | 122 | 0.32 | 0.93 | 0.28 | 1966–2020 |
| | | | | | | 1966–2020 |
| 23 | 23.1. Kurai Range, 2120 m | 90 | 0.55 | 0.97 | 0.32 | 1966–2020 |
| | *Lower part of the forest belt* | | | | | |
| 14 | 14.1. Salair Ridge, upland, 310 m | 72 | 0.24 | 0.90 | 0.25 | 1966–2020 |
| | 14.2. Salair Ridge, upland, 310 m | 71 | 0.45 | 0.96 | 0.35 | 1966–2020 |
| 15 | 15.3. Tigiretsky ridge, 590 m | 54 | 0.37 | 0.95 | 0.30 | 1966–2020 |
| 16 | Terrace of the Lebed' River, 330 m | 93 | 0.17 | 0.86 | 0.26 | 1966–2020 |
| 17 | Maima river valley, 370 m | 84 | 0.52 | 0.97 | 0.25 | 1966–2020 |
| 18 | Shore of Teletskoye Lake, 450 m | 100 | 0.27 | 0.92 | 0.31 | 1966–2020 |
| 19 | Valley of the Katun River, 490 m | 123 | 0.40 | 0.95 | 0.22 | 1966–2020 |
| 21 | Ursul basin, 1035 m | 71 | 0.76 | 0.99 | 0.25 | 1966–2020 |
| 23 | 23.2 Kurai Range, 1330 m | 69 | 0.51 | 0.97 | 0.37 | 1966–2020 |
| 24 | Chui hollow, 1760 m | 56 | 0.20 | 0.88 | 0.23 | 1966–2020 |

The larch chronology from the upper boundary of the North-Chuyskii ridge (site 22, EPS 0.97, mean sensitivity −0.28) (Table 6) shows a positive reaction to the air temperatures in March (R = 0.25), April (R = 0.28), June (R = 0.24), and September (R = 0.28), as well as a negative reaction to precipitation in September (R = −0.36) of the previous year and in February this year (R = −0.27). This chronology responds to the snow cover indicators of both the nearby (Onguday) and remote (Biysk-Zonalnaya) meteorological stations. For this chronology, there are statistically significant positive correlation coefficients with the maximum values of snow cover depth and water content for October and April at the Onguday meteorological station, and with water content for January, February, and March at the Biysk-Zonalnaya meteorological station (Table 5). At the same time, the later the stable snow cover disappears at the latter meteorological station, the greater the increase in larches; this might be due to the lack of moisture for larches in some years.

The larch chronology of the Kurai Range (site 23.1, EPS—0.96, mean sensitivity—0.30) (Table 6) responds positively to air temperatures in December–February (R = 0.30–0.34), May (R = 0.32), July (R = 0.24) and negatively to June temperatures (R = −0.28). There are no statistically significant ones with the amounts of atmospheric precipitation, but this chronology has feedback with the maximum values of depth (R = −0.36) and water content (R = −0.26) of snow cover (including for individual months—R from −0.25 to −0.61), as well as with the duration of stable snow cover (R = −0.23). At the same time, the later the snow cover is established in winter, the greater the radial growth of larches during the next year (R = 0.23) (Figure 3).

In the lower part of the forest belt, trees also react differently to climatic factors, depending on the species and geographical location. On the Salair ridge (sites 14.1 and 14.2, EPS 0.90 and 0.96, respectively, mean sensitivity—0.25 and 0.26) (Table 6), as shown by the analysis of the correlation between the fir tree-ring chronology (site 14.1) and air temperatures at the Kuzedeevo weather station, only April temperatures (R = 0.31) are statistically significant correlation coefficients. For linden trees (site 14.2), the most significant are the temperatures of May (both of the current and the previous year), to which they respond with an increase in growth (R = 0.23). Analysis of the correlation between tree-ring chronologies and precipitation totals at the Kuzedeevo weather station showed that fir trees respond positively and statistically significantly to precipitation in February (R = 0.31); precipitation in October, in contrast, has a negative impact on the growth of fir trees (R = −0.23). Linden trees (site 14.2), on the other hand, respond statistically significantly to an increase in precipitation only in February (R = 0.23), March (0.41), and June (R = 0.24). Trees in this area do not significantly react to snow cover indicators, except for the maximum values of the depth and water reserve of the snow cover in October and November of the previous year (Table 5).

In other areas of the lower part of the forest belt, trees of different species either do not respond to the maximum depth and water content of snow cover (plots 16, 17, 24), or they react positively (plots 15.3, 18, 19, 21, 23.2) (Table 5, Figure 2). Most often, significant coefficients are typical for the first and last months of winter (R = 0.23–0.68). The tree-ring chronologies of the lower part of the forest belt more often react positively to the early melting of the snow cover. However, in some cases (site 23.2), this reaction is the opposite (Figure 3). A similar situation is noted for the duration of stable snow cover.

## 4. Conclusions

Research revealed that snow cover is not the principal factor limiting the tree-ring growth of woody plants, even under extreme conditions. Its impact on woody plants depends on a complex of factors, including local ones.

The maximum depth and water content of the snow cover are the most important characteristics of the snow cover for the growth of woody plants in the forest-tundra and dry steppe, as well as on the upper forest boundary (especially in the Polar Urals). In the forest-tundra, these indicators control the degree of soil freezing and, probably, the rate of their warming up in spring and the beginning of tree growth. In the dry steppe, an increase in the depth and water content of the snow cover contributes to better soil moisture, which ensures better tree growth. The impact of these indicators is somewhat enhanced in the forest-tundra on the lower parts of the slopes and in the dry steppe on the uplands and crest tops. The trend of increasing snow reserves in these areas contributes to an increase in the rate of radial growth of trees. At the same time, in humid locations of the dry steppe with a high level of standing groundwater, an increase in these indicators leads to a slowdown in the growth of the studied tree species. The limiting value of the depth and water content of snow cover for the radial growth of trees in the northern and middle taiga and southern forest-steppe is less than in the forest-tundra and northern forest-steppe.

The dates of the disappearance of stable snow cover are more significant in the southern regions (southern and, especially, northern forest-steppe and in the lower part of the Altai forest belt) than in the northern regions (forest-tundra and northern taiga). At the

same time, an earlier melting of the snow cover contributes to the increase in the radial growth of trees in the forest-steppe, and later in the lower parts of the forest belt of the central regions of the Altai.

The dates of the establishment of stable snow cover are more important for tree growth compared to the dates of disappearance, especially in the forest-tundra and forest-steppe. In these areas, the later establishment of snow cover ensures a greater radial growth of trees, especially in the next year.

The value of the duration of the period with stable snow cover for tree growth is higher in the southern regions. At the same time, in the forest-steppe, the longer the occurrence of a stable snow cover, the less the growth of trees. A similar situation is noted in the lower part of the Altai forest belt.

**Author Contributions:** Conceptualization, N.I.B.; Methodology, N.I.B.; Formal Analysis, N.I.B., N.V.R. and A.A.S.; Investigation, N.I.B., N.V.R. and A.A.S.; Resources, N.I.B.; Data Curation, N.V.R.; Writing—Original Draft Preparation, N.I.B., N.V.R. and A.A.S.; Writing—Review and Editing, N.I.B.; Visualization, N.I.B. and A.A.S. All authors have read and agreed to the published version of the manuscript.

**Funding:** The study was supported by the grant of the Russian Science Foundation No. 22-27-00268 "Reconstruction of the long-term dynamics of nival-glacial phenomena in the contrasting landscape conditions of Altai according to tree-ring indication", https://rscf.ru/project/22-27-00268/ (20 July 2023).

**Data Availability Statement:** The data that support the findings of this study are not openly available and are available from the corresponding author upon reasonable request.

**Acknowledgments:** We are grateful to Richard Bland for his great assistance in the English proof-reading of the work.

**Conflicts of Interest:** The authors declare no conflict of interest. The funders had no role in the design of the study; in the collection, analyses, or interpretation of data; in the writing of the manuscript; or in the decision to publish the results.

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
