# Peer review of "Response of the Radial Growth of Woody Plants in the West Siberian Plain and Adjacent Mountainous Territories to the Characteristics of the Snow Cover"

_forests, doi:10.3390/f14081690_

Round 1

Reviewer 1 Report

This research represents a lot of work, starting with many field sites scattered across a large area and encompassing multiple tree-ring collections at each site.  Impressive.  After all is said and done, the conclusion might seem disappointing given the emphasis of the work: snow cover is not the principal factor limiting the tree-ring growth of woody plants (lines 369-370).  Oh well, but negative findings are still publishable.

Methods: Not much detail is given on the dendrochronological analysis.  Typically in dendroclimate papers, an index of site strength across trees is given.  EPS is not quite that, but one index is site Rbar, which is given in ARSTAN outputs.  Can that be included in a table?  As it stands currently, there’s little to no indication of how well trees crossdate with each other within sites, a fundamental attribute of interest in dendrochronology.  Was COFECHA applied here?

Two word issues:

• Thickness is used to describe snow cover, which I guess is understandable but I believe in English the word depth would be more standard.

• Route?  In the caption for Figure 2, route is used in a way that I’ve not seen before and don’t follow.  I’m not even sure what alternative word to suggest.

Author Response

Methods: Not much detail is given on the dendrochronological analysis.  Typically in dendroclimate papers, an index of site strength across trees is given.  EPS is not quite that, but one index is site Rbar, which is given in ARSTAN outputs.  Can that be included in a table?  As it stands currently, there’s little to no indication of how well trees crossdate with each other within sites, a fundamental attribute of interest in dendrochronology.  Was COFECHA applied here?

Added information about the dendrochronological method and Pbar tree-ring chronologies (tables).

Comments on the Quality of English Language

Two word issues:

  • Thickness is used to describe snow cover, which I guess is understandable but I believe in English the word depth would be more standard.

Accepted, corrected

  • Route? In the caption for Figure 2, route is used in a way that I’ve not seen before and don’t follow. I’m not even sure what alternative word to suggest.

corrected  - snow course survey

Reviewer 2 Report

Interesting work on the impact of snow cover on the radial growth of trees growing in different zones of the West Siberian Plain. The introduction of the work directly refers to the topic. The purpose of the work was actually set.  My main comments concern the section of the Materials & Method. This section should be improved.  The climate database is described very precisely, while the information about the growth data and dendrochronological procedures is insufficient. Information about dendrochronological analyzes is contained in one sentence. (lines 66-67: Standardization and generalization of dendrochronological series was performed in the ASTAN program).

In the next sentence, the authors refer only to their works in which "Some results of data analysis of dendrochronological samples have been published [3,4,19,21". In the Results and Discussion section, the authors present the results of the analyses, which unfortunately have not been described in the methods.

 Therefore, the "Methods" section should be supplemented with information on data transformation, the method of building chronology, their statistics, and climate-growth analyses.

1. How old were the trees from which the samples were taken?

What was the length of the chronology for each species (please specify years)

2. What methods/procedures were used in the construction of the chronologies

3. What versions of chronology were used in the analyses? – raw/standardized/residual

4. What methods were used to determine the relationships between climate and growth?

5. In the Method section, there is no information about the chronology statistics provided in the Results section - e.g. EPS, sensitivity,

6. What software was used for calculations?

 The Results and Discussion section seems to be well conducted, but the assessment of the correctness of this section can be made only after the authors present the methodology of dendrochronological analysis.

 Minor comments:

1.           Descriptions of the axes in Figures 2 and 3 - are hard to read

2.           What do the blanks in Table 3 mean, is it missing data? If yes, please write it in the explanation of the Table

3.           Line 179 -  the thickness of annual rings – change  on „width tree-ring” 

 Best regards,

there are some grammatical errors, please check the work, e.g. in Grammarly

e.g. line (lines 66-67: Standardization and generalization of dendrochronological series was performed in the ASTAN program- should be were ).

Terminology errors - eg

Coefficients of correlation between the thickness of annual rings of woody plants in the studied sites (Figure 1, Tables 1 and 2) with the maximum thickness

  it should be "width"

Author Response

Added information about the dendrochronological method and about Rbar, EPS, average sensitivity of tree-ring chronologies and average tree age (tables).

Line 179 -  the thickness of annual rings – change  on „width tree-ring”and other terminological errors - corrected.

Axes of figures are read badly only in PDF format. However, the drawings are of high resolution and there will be no such problems with a separate download.

Reviewer 3 Report

The paper analyzed the relationship between the tree ring index and snow cover with 24 plots. The research included a number of plot sites whose conclusion may comprehensively display the influences of snow cover to the tree growth. Based on the following reasons, I suggest a major revision.

1.       The biggest novelty is the number of plots involved. The readers may want to know how many plot data are never used in the previous publications. If most of the plots have been used, it may spoil the novelty of the paper.

2.       The authors can tell us what the threshold of EPS for the involved plots is. In the paper, I can see some EPS for specific plots but not the overall EPS for all plots.

3.       I just assume the snow cover can also be affected by the temperature and precipitation. Maybe the author can add some figures or tables about the correlations between tree ring index and temperature (or precipitation)

4.       I just make sure that the tree ring width in the paper indicates the tree ring index but not the raw tree ring width.

5.       In the discussion part, the authors list detailed results. But I want to find some clear take-home message like how elevation, aspect (south/north), or tree species influences the correlation between snow and radical growth. I suggest that the discussion can be consisted of a few parts like elevation, aspect, and tree species. I just want the discussion will be easy to read.

Moderate editing of English language required.

Author Response

Responses to comments

  1. The biggest novelty is the number of plots involved. The readers may want to know how many plot data are never used in the previous publications. If most of the plots have been used, it may spoil the novelty of the paper.

Most of the sites have not previously been used to analyze the effect of snow cover characteristics on the radial growth.

  1. The authors can tell us what the threshold of EPS for the involved plots is. In the paper, I can see some EPS for specific plots but not the overall EPS for all plots.

Added tables with this information

  1. I just assume the snow cover can also be affected by the temperature and precipitation. Maybe the author can add some figures or tables about the correlations between tree ring index and temperature (or precipitation)

Information about the reaction tree-ring chronologies is available in the text. References are also made to our works, where this information is available. Adding pictures will make the job cumbersome.

  1. I just make sure that the tree ring width in the paper indicates the tree ring index but not the raw tree ring width.

For dendroclimatic analysis, width indices were used, since the construction of local chronologies requires a standardization procedure

  1. In the discussion part, the authors list detailed results. But I want to find some clear take-home message like how elevation, aspect (south/north), or tree species influences the correlation between snow and radical growth. I suggest that the discussion can be consisted of a few parts like elevation, aspect, and tree species. I just want the discussion will be easy to read.

This is difficult to do because the territory is large and highly differentiated in physical and geographical terms. Even within the Altai, the conditions differ greatly in its regions. Therefore, comparison in height within different areas will be incorrect. There are also difficulties in conducting such a study within the same ridge, since the upper and lower forest boundaries are often represented by different tree species.

Reviewer 4 Report

General remarks of the reviewer

The article is an interesting study on the reaction of the width of tree rings to meteorological factors and snow cover.

The following chapters requires some clarification:

1.Introduction:

The literature review is relatively poor, in particular, please complete with the most important publications from other regions of the world. At the end of the introduction, please clearly articulate the research goal and working hypotheses.

2.Materials and Methods:

The description of the empirical material should strongly emphasize its great diversity, species with different environmental requirements, non-uniform periods of comparison with meteorological conditions and large differences in the distance of meteorological stations from the study plots. This will have consequences when analyzing the research results.

3.Results and Discussion:

For transparency, I suggest separating chapter 3. Results and Discussion into 3. Results covering own research results and 4. Discussion their confrontation with research by other authors.

Wouldn't it be better to use the response function and clear visualization of these dependencies using a table with a color scale to analyze the reaction of the width of tree rings to meteorological factors and snow cover?

4.Conclusions:

Conclusions should be redrafted in the direction of cognitive and application use of research results. In their current form, they are a shortened version of the result and discussion parts.

Technical Notes

Make the Figures  and Tables clearer if possible.

Use italics for latin names of species throughout the text.

Eliminate logical errors and typos.

The description of the literature item needs to be corrected as required by the publisher: articles, books and other sources - italics of journal titles, year in bold, correct pages of journals and the access link and date of access in English. According to MDPI standard.

Details in the attached manuscript.

Minor editing of English language required.

Author Response

responses to comments

1.Introduction:

The literature review is relatively poor, in particular, please complete with the most important publications from other regions of the world. At the end of the introduction, please clearly articulate the research goal and working hypotheses.

There are works on Europe, North America, Asia. This information is not available for Africa and Australia. We are also not aware of such works in South America. If the reviewer can cite such articles, then we will consider them with pleasure.

 2.Materials and Methods:

The description of the empirical material should strongly emphasize its great diversity, species with different environmental requirements, non-uniform periods of comparison with meteorological conditions and large differences in the distance of meteorological stations from the study plots. This will have consequences when analyzing the research results.

This information is in the works. Added tables have additional information

3.Results and Discussion:

For transparency, I suggest separating chapter 3. Results and Discussion into 3. Results covering own research results and 4. Discussion their confrontation with research by other authors.

The requirements for Lesa magazine authors do not state that a separate Discussion section is mandatory. It is possible to combine it with the "Results" section. In this paper, it seems to us more convenient to discuss immediately after the presentation of the results. For this case, we have a joint section "Results and Discussion". The meaning of the paper is still very clear.

Wouldn't it be better to use the response function and clear visualization of these dependencies using a table with a color scale to analyze the reaction of the width of tree rings to meteorological factors and snow cover?

Maybe. However, at this stage of the research, we used only the calculations of the Pearson correlation coefficients

4.Conclusions:

Conclusions should be redrafted in the direction of cognitive and application use of research results. In their current form, they are a shortened version of the result and discussion parts.

I cannot agree with the evaluation of the conclusions. In the "Results and Discussion" section for each site, the influence of various factors on the radial growth of trees is considered. In the “Conclusions” section, conclusions are drawn about how the influence of one or another indicator of snow cover changes on the radial growth of trees throughout the study area. That is, we see territorial differentiation of the significance of a particular indicator of snow cover

Technical Notes

Make the Figures  and Tables clearer if possible.

The drawings have a high resolution and when they are downloaded separately, there will be no such problems as in pdf format

Use italics for latin names of species throughout the text.

Corrected

Eliminate logical errors and typos.

Corrected

The description of the literature item needs to be corrected as required by the publisher: articles, books and other sources - italics of journal titles, year in bold, correct pages of journals and the access link and date of access in English. According to MDPI standard.

Corrected

Round 2

Reviewer 3 Report

The authors answered me most of the questions. But one comment has not response. How many plots in this research are the first time to show their data which have not been used in the previous research? 

Author Response

Point 1: But one comment has not response. How many plots in this research are the first time to show their data which have not been used in the previous research?

Response 1: Approximately 20% of the chronologies were previously used to analyze the relationship between the width of annual rings and the maximum snow cover indices for the winter period. Graphically, these results have never been published. Also, chronological data were not used to analyze the relationship between the width of annual rings and snow cover indicators for individual months.

Reviewer 4 Report

The Authors responded to the main comments and added important information about the research (including data in tables).

Correct the previously indicated irregularities:

line 209 - in the title of Table 3, change the year 1920 to 2020,

use italics for latin names of species throughout the text (also in tables).

After making these minor changes, the article can be published.

Minor editing of English language required.

Author Response

Point 1:  line 209 - in the title of Table 3, change the year 1920 to 2020,

use italics for latin names of species throughout the text (also in tables).

Response 1:.Corrected

Point 2:  Minor editing of English language required.

Response 2: English is verified by a native speaker whose name is listed in the Acknowledgments
